# Factors that influence market participation among traditional beef cattle farmers in the Meatu District of Simiyu Region, Tanzania

**Cornel Anyisile Kibona** [1,2], **Zhang Yuejie** [1] *

**1** College of Economics and Management, Jilin Agricultural University, Changchun, Jilin, China,
**2** Department of Agricultural Economics and Finance, Mwalimu Julius. K. Nyerere University of Agriculture and Technology, Musoma, Tanzania

* zhang-yj@163.com

**Data Availability Statement:** All relevant data are within the manuscript and its Supporting Information files.

**Funding:** The research is supported and funded by the National Beef-cattle Industrial Technology

## Abstract

Market participation among beef cattle farmers is key to ensuring better income, food security, and sustainable beef supply. Farmers in the traditional beef cattle sector, nevertheless, are well known for their low market participation. This study, thus, sought to examine factors that influenced market participation among beef cattle farmers in the Meatu district of Simiyu region, Tanzania. The study randomly selected a sample size of 393 farmers. The cross-sectional data collected through interviews and questionnaires were analyzed using descriptive statistics and ordinary least squares (OLS) regression models. The descriptive analyses showed that the average age of the interviewees was 53.73 years with a family size of 13.11. On average, the respondents had about 24.14 years of farming experience. Most of the respondents (67.9%) had primary education. Among the respondents, about 61.3%, 4.6%, and 32.6% had access to market information, farm credits, and veterinary services, respectively. The average cattle herd size was 53.46 heads. About 90.1% of farmers had participated in the markets by selling an average of 5 heads each, per year. This study has revealed low volume of sales, low education levels, and poor access to credits and veterinary services as the major factors that limit market participation. Furthermore, econometric results show that the key factors that significantly influenced farmers to increase the volume of beef cattle sales in the market at $P < 0.05$ included price, herd size, off-farm income, distance to market, age of the farmer, and cattle fattening. Moreover, membership in cooperatives, access to market information, and farming experience also had significant influence at $P < 0.05$. This study recommends establishment of strategic cooperatives to function not only as a communication channel for farm credits, price, market information, and training on commercial farming, but also to assist farmers in selection of profitable markets.

## 1. Introduction

Increase in beef cattle sales through market participation has become an important aspect in commercialization of the traditional beef cattle sector in rural areas [1].

System and Industrial Economy Research Project under the Ministry of Agriculture in the People's Republic of China (PRC) (CARS-37). The funders had no role in study design, data collection and analysis,decision to publish,or preparation of the manuscript.

**Competing interests:** The authors have declared that no competing interests exist.

Globally, it is estimated that over 600 million people keep beef cattle, and nearly 75% lives in rural areas [2]. Beef cattle production in Tanzania is one of the major agricultural production sectors. It contributes to over 50% of the beef cattle farmer's household income, 5.9% to the national GDP, export earnings, and a great source of employment [3–6]. Despite its potential for economic development, the traditional beef cattle sector has thinly been developed, partly due to the limited commercialization (market participation) [7]. Various factors are attributed to the limited market participation among traditional beef cattle farmers in Tanzania. These factors include limited access to market and its opportunities [8].

Tanzania is ranked as the third (3rd) leading producer of beef cattle in Africa and 11th in the world. The country is estimated to have a beef cattle population of over 34.5 million with a 2.8% annual growth rate [1, 5, 6]. Beef cattle farming in Tanzania is, to a great extent, traditional. Available data indicate that 94% of total beef cattle herds is predominantly produced under the traditional beef cattle farming (a free-range production system), whereas, only 6% is under commercial beef cattle ranching [4, 9]. The potential for growth of the beef cattle in Tanzania is high since the country has favourable conditions and vast land, which can support growth of the sector. Approximately, Tanzania has 60 million hectares of rangelands suitable for grazing [3].

Given that 94% of beef cattle herds in Tanzania is produced under the traditional beef cattle sector which supplies 95% of the country's beef meat [3, 4], the government strives to commercialize the sector. The effort intended to ensure that the sector effectively promotes household food security and income, as well as responding to increasing demand from domestic and international meat markets [10, 11]. The efforts undertaken to commercialize the sector include; linking farmers to profitable markets, subsidizing inputs, and enabling farmers' access to credit and veterinary services [12, 13]. In addition, the government encourages farmers to reduce large cattle herds, settle on land allocated for grazing, and fatten beef cattle before selling (value addition) [10, 12, 13].

Beef cattle commercialization typically leads to increased diversity of marketed commodities and specialization. This encourages farmers to produce high quality beef cattle, thereby increase their incomes [14]. Market participation is related to commercialization: it refers to the gradual shift from traditional to commercial beef cattle farming. It also refers to the farmers' act of entering the beef cattle market to exchange their products for cash [15]. Generally, market participation ensures; a continuous supply of beef cattle to markets, farmers' better income, food security, and poverty reduction.

Despite the government effort to commercialize the traditional beef cattle sector, most traditional beef cattle farmers continue to live in poverty [10, 11]. The country, thus has continued to import quality beef meat (over 700 metric tons per year), meanwhile, the existing beef meat processing abattoirs have been operating at less than 50% of their operational capacities [3, 4, 16, 17]. This is quite an indication of a low supply of beef cattle in the market. It has been observed that wealth protection and prestige that is perceived by cattle farmers after accumulating huge beef cattle herds exceed market incentives, thus jeopardizing the integration of beef cattle farmers into organized markets [18]. The traditional beef cattle sector in Tanzania still has huge potential to boost income from and supply of beef cattle, both, domestically and internationally: farmers' market participation is most likely one of the best means to this end.

According to the Tanzania Livestock Master Plan (TLMP) [9], projected beef meat demand-supply gap by 2031/32 is estimated to be 1.7 million tons. The projected high demand of beef meat is driven by an increasing human population, particularly in developing countries, income growth, urbanization, growing tourist industry, and income elasticity of demand.

This expansion provides potential opportunities to beef cattle farmers to increase earn more income through market participation.

Recent estimates indicate that, due to low market participation among traditional beef cattle farmers in Tanzania, by the year 2021/22, contribution of the traditional beef cattle sector to the national red meat production will decrease by 8.43%, whereas, that of commercial beef cattle ranching will increase by 85.94% [9]. The commercial beef cattle ranching system is more market-oriented than the traditional one. More efforts should be put to transform traditional beef cattle farmers from subsistence farming to market-oriented; beef cattle farmers need to perceive beef cattle farming as a business. This may encourage beef cattle farmers to set aside larger volumes of beef cattle for sale each year [16].

This study, thus, sought to examine factors that influenced market participation among traditional beef cattle farmers in Tanzania: this is essential for establishing a sustainable development policy framework for maximizing rural economic growth and sustainable beef cattle supply to domestic and international markets [18].

## 2. Materials and methods

### 2.1. Description of the study site

This study was conducted in the Meatu District of the Simiyu Region. Located in northern Tanzania, southeast of Lake Victoria, Simiyu Region, particularly the Meatu District, is dominated by traditional beef cattle farming as its major economic activity. Simiyu Region is among the regions with high beef cattle populations, with a significant contribution to the national beef cattle herd stock. With a population of 1,584,157 people, the region is estimated to hold a total of 1,512,911 beef cattle, and covers an area of 25,212 square kilometres. The study district has a population of 299,619 people, and it is estimated to hold a total of 495,890 beef cattle. Rainfal in Meatu District ranges between 600 mm and 900 mm per year and temperature ranges from 18 $^0$C to 31 $^0$C. Generally, grazing land occupies 80% of the total area and the remaining 20% is used for agricultural production and settlements [5, 19].

### 2.2. Ethical considerations

This study was first approved by the Jilin Agricultural University Graduate Research Ethics Committee in China. It was then submitted to and approved by the Ministry of Livestock and Fisheries (MLF) with reference number (AB.16/2020/01). During the data collection process, all ethical considerations were dealt with accordingly; first, the participants' consent was obtained verbally, and then the detailed information in the consent form was explained to all the participants. The participants were then allowed to fill in and sign the forms as proof of their consent to participate in the study before their actual participation. All the participants were informed of their right to decline their participation anytime they felt so. The Ward Executive Officer (WEO) witnessed and approved the consent.

### 2.3. Sampling procedures

The study applied a multi-stage stratified sampling technique to select respondents among beef cattle farmers. Selection of respondents at different stages involved purposive and randomised sampling. Stratified random sampling creates stratification based on members who share similar attributes [20]. Strata in this study were made of the five major beef cattle producing regions and their districts. One region (Simiyu) was then randomly selected from the five regions. Similarly, one district (Meatu) was purposely

selected among the five districts since it is the leading beef cattle producing district in the region. In the study district, three villages; Nkoma, Mwambegwa, and Mwambiti were randomly selected. This study targeted traditional beef cattle farmers (N = 24,139) and applied Slovin's formula to determine a randomly selected sample size of 393 respondents [21] as;

$$n = \frac{N}{1 + Ne^2} = \frac{24,139}{1 + 24,139\,(0.05)^2} = 393.48 \approx 393 \tag{1}$$

Whereby $N$ is the targeted population size, $n$ is a sample size, and $e$ is the error tolerance level. The number of respondents selected from each village (stratum) was determined by utilizing the percentage proportion (see Table 1).

## 2.4. Data collection

The cross-sectional survey was used to collect primary data using structured questionnaire and interview methods. The structured questionnaires captured three sections. Section I included the socioeconomic characteristics of beef cattle farmers, section II was designated for beef cattle production information, and section III for beef cattle sales (market participation) information.

## 2.5. Analytical models

This study applied both descriptive statistic and econometric models. Firstly, the socioeconomic characteristics of beef cattle farmers, beef cattle marketing activities, together with public service related factors were examined using descriptive statistics which involved percentages, frequencies, means, and standard deviation.

Secondly, in econometric analysis, ordinary least squares (OLS) multiple linear regression model was applied to determine factors that influenced beef cattle farmers intention to increase the volume of beef cattle sales in the market. The OLS is a mathematical modelling method that can be used to explain the relationship between a continuous dependent variable (volume of beef cattle sold) and multiple independent variables [22]. The actual OLS model used was as follows:

$$Y_i = \beta_0 + \beta_1 X_1 + \beta_2 X_2 + \beta_3 X_3 + \ldots\ldots\ldots + \beta_n X_n + \varepsilon_i \tag{2}$$

Where

$Y_i$ denotes the number of beef cattle sold, $\beta_0$ is an intercept or constant, $\beta_{1,\ldots,}\beta_n$ are the coefficients to be estimated, and $X_1,\ldots,$ and $X_n$ represent the vectors of the explanatory variables and $\varepsilon_i$ the error term.

**Table 1. Sample size.**

| District | Villages | Population | Percentage Proportion | Sample |
|---|---|---|---|---|
| Meatu | Nkoma | 8,006 | 33.17 | 130 |
| | Mwambiti | 7,943 | 32.91 | 129 |
| | Mwambegwa | 8,190 | 33.93 | 133 |
| **Total** | | **24,139** | **100** | **393** |

The truncated OLS regression model for factors determining the volume of beef cattle sold was specified as:

$$Number\ of\ beef\ cattle\ sold$$
$$= \beta_0 + \beta_1\ Age + \beta_2\ Education + \beta_3\ Acces\ to\ market\ information$$
$$+ \beta_4\ Distance\ to\ market + \beta_5\ Household\ size + \beta_6\ Beef\ cattle\ herd\ size$$
$$+ \beta_7\ Farming\ experience + \beta_8\ offFarm\ income$$
$$+ \beta_9\ Access\ to\ veterinary\ service + \beta_{10}\ Access\ to\ credits \tag{3}$$
$$+ \beta_{11}\ Grazing\ land\ owned\ + \beta_{12}\ Price\ of\ beef\ cattle$$
$$+ \beta_{13}\ Beef\ cattle\ fattening\ practice + \beta_{14}\ Cooperative\ membership$$
$$+ +\beta_{15}\ Cows + \beta_{16}\ Bulls + \beta_{17}\ Oxen + \beta_{18}\ Heifers + \beta_{19}\ Localbreads + \varepsilon$$

Table 2 below shows the hypothesized sign effects of the independent variables used in the OLS multiple regression model.

## 2.6. Conceptual framework

To commercialize the traditional beef cattle sector in Tanzania for a sustainable supply of beef cattle to the market and poverty reduction among beef cattle farmers, this study conceptualized that traditional beef cattle commercialization is dependent on the relationship between increased beef cattle sales (extent of market participation) and their determinants.

The relationship between the determinant (independent) variables and the dependent variable included in the study, together with the externalities, are illustrated in Fig 1. Thus, an increased volume of beef cattle sales in the market (dependent) was assumed to be influenced by farmers' age, education level, beef cattle herd size, cattle farming experience, cooperative membership, grazing land owned, access to market information, access to credits, and access

**Table 2. Independent variables and their expected effects on the volume of beef cattle sold by beef cattle farmers.**

| Variables | Variable type | Measurements | Hypothesized signs |
|---|---|---|---|
| Age of a farmers | Continuous | Age in years | ± |
| Education level | Continuous | Years of schooling | + |
| Access to market information | Dummy | If 0 = No,1 = Yes | + |
| Distance from home to market | Continuous | Total kilometres | - |
| Household size | Continuous | Number of members | ± |
| Beef cattle herd size | Continuous | Number of beef cattle | + |
| Beef cattle farming experience | Continuous | In years | + |
| Off-farm income | Continuous | In Tanzanian shillings | ± |
| Access to veterinary services | Dummy | If 0 = No,1 = Yes | + |
| Access to credits | Dummy | If 0 = No,1 = Yes | + |
| Grazing land owned | Continuous | In hectare | + |
| Price of beef cattle | Continuous | In Tanzanian shillings | + |
| Beef cattle fattening practice | Dummy | If 0 = No,1 = Yes | + |
| Cooperative membership | Dummy | If 0 = No,1 = Yes | + |
| Cows owned | Continuous | Number of cows | ± |
| Bulls owned | Continuous | Number of bulls | + |
| Steers-Oxen owned | Continuous | Number of oxen | - |
| Heifers owned | Continuous | Number of heifers | ± |
| Local breed beef cattle | Continuous | Number of local breeds | - |

to veterinary services. Moreover, other factors assumed to influence the volume of beef cattle sales in the market included distance to market, price of beef cattle, off-farm income, household size, beef cattle fattening practice, number of cows owned, number of bulls owned, and the local beef cattle breed.

The conceptual framework also shows interaction of some externalities supporting increased volumes of beef cattle sales. Based on the above factors, the expected outcomes are: poverty reduction, increase in beef meat production and beef cattle supply to both domestic and international markets, which finally leads to the national economic growth and foreign direct investment (FDI) in the beef cattle sector. The conceptual framework in Fig 1 underpins the adoption and development of empirical analysis.

## 3. Results and discussions

### 3.1. Descriptive analysis

**3.1.1. Comparison of the socioeconomic characteristics of market and non-market participants based on the average scores of continuous variables.**   Results in Table 3 show that among the 393 sampled beef cattle farmers, about 90.1% (354) had participated in the beef cattle market by selling an average of 5 beef cattle per year, while 9.9% (39) did not engage in beef cattle marketing at all. This indicates that the majority of beef cattle farmers are dependent on

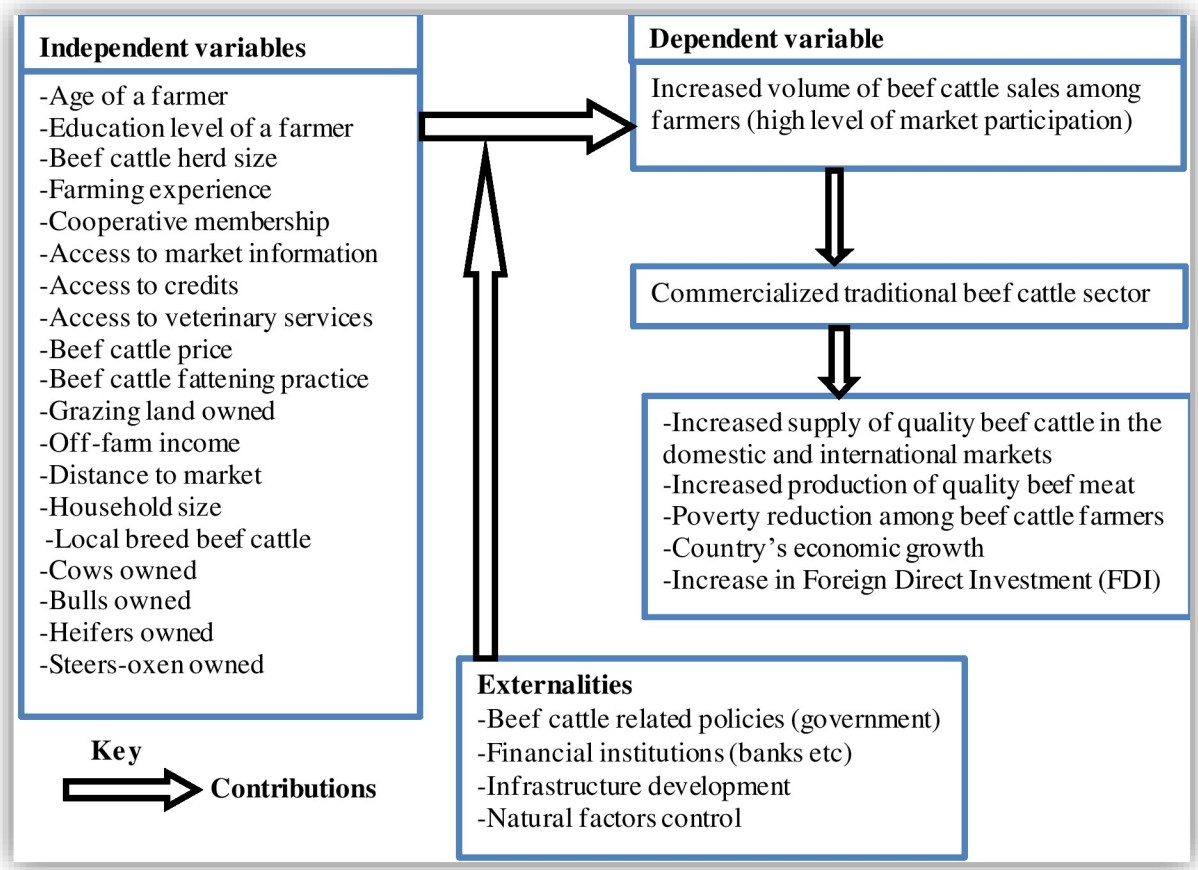

**Fig 1. Conceptual framework.**

**Table 3. Comparison of socioeconomic characteristics of market and non-market participants based on the average scores of continuous variables.**

| Variables | Overall (N = 393) | Market participants (N = 354) | Non-market participants (N = 39) |
|---|---|---|---|
| Age of a farmer | 53.73 | 54.52 | 46.56 |
| Household size | 13.11 | 13.41 | 10.44 |
| Beef cattle herd size | 53.46 | 54.46 | 44.31 |
| Farming experience | 24.14 | 25.04 | 18.74 |
| Off-farm income(US$) | 1,169 | 1,228 | 626 |
| Grazing land owned (ha) | 38.35 | 39.88 | 24.38 |
| Cows owned | 23.93 | 24.32 | 20.36 |
| Bulls owned | 11.43 | 11.75 | 8.59 |
| Steers-Oxen owned | 8.89 | 8.83 | 9.49 |
| Heifers owned | 9.59 | 10.00 | 5.87 |

income generated from beef cattle sales to secure their financial situations. The average age of market and non-market participants was 54.52 and 46.56 years, respectively. This indicates that both market and non-market participants were in the active age of the labour force, which is important in the adoption of beef cattle production technologies and a willingness to engage in beef cattle marketing. The average household size among market participants was 13.41, which is higher than the 10.44 obtained for non-market participants. This indicates a higher labour force potential for beef cattle production and marketing activities among market participants. It also shows that market participants had 25.04 years of farming experience, which is higher than the 18.74 years obtained for non-market participants. Beef cattle farming experience increases beef cattle productivity through acquisition of skills and knowledge, thereby increasing the farmers' probability to participate in the market [18, 23]. Furthermore, the results revealed that the market and non-market participants had an average of 54.46 and 44.3 beef cattle herd size, respectively. These results indicate that the larger the beef cattle herd size, the more the beef cattle farmers' motivation to participate in the markets. The study results also revealed that the average off-farm income among market participants was 1,228 US$ per year. This was higher than the 626 US$ obtained for non-market participants. Apparently, off-farm income increases farm productivity if reinvested in beef cattle production, thereby increasing the chance of beef cattle farmers participating in the market. The mean grazing land owned by market participants was 39.88 ha, while the corresponding value for the non-market participant was 24.38 ha. Grazing land availability is important in beef cattle productivity, which enhances availability of the beef cattle for sale.

The study findings also show that the beef cattle flock characteristics such as cows, bulls, and heifers were higher on average among market participants. These factors are concerned with beef cattle herd dynamics marked with increased beef cattle productivity, hence promoting market participation. The steers-oxen average was found to be higher among non-market participants. Steers-oxen are mostly used as a source of draught animal power in rural areas, which reduces beef cattle surplus to be offered for sale, thereby reducing the opportunity to participate in the market.

**3.1.2. Comparison of socioeconomic characteristics of market and non-market participants based on frequency and percentage scores of categorical variables.** Results in Table 4 revealed that traditional beef cattle farming was dominated only by men. This indicates gender disparity: females should be encouraged to engage in beef cattle farming. Women are also responsible for farming and food processing, as men migrate from rural to urban centres in search for employment [15]. Analysis further revealed that 68.9% and 59% of the market and non-market participants, respectively, had primary education, while only 2.3% and 20.8%

**Table 4. Comparison of socioeconomic characteristics between market and non-market participants based on frequency and percentage scores of categorical variables.**

| Variables | Overall (N = 393) | | Market participants (N = 354) | | Non-market participants (N = 39) | |
|---|---|---|---|---|---|---|
| | Frequency | Percentage | Frequency | Percentage | Frequency | Percentage |
| **Gender of a beef cattle farmer** | | | | | | |
| Male | 393 | 100 | 354 | 100.0 | 39 | 100.0 |
| **Education level** | | | | | | |
| No education | 110 | 28.0 | 102 | 28.8 | 8 | 20.5 |
| Primary ed. | 267 | 67.9 | 244 | 68.9 | 23 | 59.0 |
| Secondary ed. | 16 | 4.1 | 8 | 2.3 | 8 | 20.8 |
| **Access to market information** | | | | | | |
| Yes | 241 | 61.3 | 226 | 63.8 | 15 | 38.5 |
| No | 152 | 38.7 | 128 | 36.2 | 24 | 61.5 |
| **Access to credits** | | | | | | |
| Yes | 18 | 4.6 | 16 | 4.5 | 2 | 5.1 |
| No | 375 | 95.4 | 338 | 95.5 | 37 | 94.9 |
| **Access to veterinary services** | | | | | | |
| Yes | 128 | 32.6 | 112 | 31.6 | 16 | 41.0 |
| No | 265 | 67.4 | 242 | 68.4 | 23 | 59.0 |
| **Membership to Cooperatives** | | | | | | |
| Yes | 58 | 14.8 | 53 | 15.0 | 5 | 12.8 |
| No | 335 | 85.2 | 301 | 85.0 | 34 | 87.2 |
| **Practicing beef cattle fattening(value addition) before sale** | | | | | | |
| Yes | 24 | 6.1 | 24 | 6.8 | 0 | 0.0 |
| No | 369 | 93.9 | 330 | 93.2 | 39 | 100 |

of the market and non-market participants, respectively, had secondary education. The remaining 28.8% and 20.5% of the market and non-market participants, respectively, had no formal education. These findings revealed that most farmers had low level of education. Farmers should be provided with tailor-made training and education to promote the development of the traditional beef cattle sector. Education improves one's ability to observe product quality, effectively negotiate price, and access available market information. People with high levels of education should be encouraged to invest in beef cattle production to boost quality production and marketing.

Findings in this study also show that 63.8% and 38.5% of market and non-market participants, respectively, had access to market information. This may suggest that market participants were more exposed to market information. Obtaining accurate market information is an essential factor for the farmers to participate in the market. Furthermore, this study has shown that only 4.5% and 5.1% of market and non-market participant, respectively, had access to farm credit. This implies that both market and non-market participants had poor access to farm credits. Beef cattle farmers need suitable and convenient arrangements as well as assistance in establishment of a marketing system for securing farm credit. Farm credit is important for investing in beef cattle production and marketing activities, thus boosting beef cattle productivity, which increases the tendency to enter the market [18]. Regarding access to veterinary services, results show that 31.6% and 41% of market and non-market participants, respectively, had access to veterinary services. This also indicates that farmers had limited access to veterinary services. Access to veterinary services should be improved through reducing the cost of acquiring consultations necessary for improved beef cattle production.

Findings in this study also revealed that involvement in cooperative activities was limited among, both, market and non-market participants. Results show that only 15% and 12.8% of market and non-market participants, respectively, were engaged in farmer's cooperatives. Cooperatives help farmers mobilize resources, share market information, improve their bargaining power, promote their services, and reduce cost through economies of scale [18]. Incentives should be set forth to attract beef cattle farmers to join cooperatives.

Findings in this study further revealed that only 6.8% of market participants added value (fattening) to their beef cattle before sale, whereas, none among the non-market participants did. The findings indicate low beef cattle value addition among farmers. In Tanzania, fattening beef cattle has been designated as one of several ways to increase beef cattle productivity and quality [24]. Beef cattle of a lower quality are fed with cotton husks, corn bran, or cottonseed cakes for three to four months to improve their quality by gaining weight, and then sell them at high prices [24]. Incentives to encourage beef cattle farmers to practice fattening are imperative. Generally, these findings suggest that higher mean values of variables among market participants positively influenced market participation.

## 3.2. Econometric analysis

### 3.2.1. Factors influencing the volume of beef cattle sold in the market by beef cattle farmers.
Table 5 shows the ordinary least squares (OLS) multiple linear regression output on the factors that influenced the volume of beef cattle sold in the market by beef cattle farmers. The $R^2$ value from the OLS multiple regression analysis was 0.792. This shows that the

**Table 5. Ordinary Least Squares (OLS) multiple linear regression estimates on the factors determining the volume of beef cattle sold in the market by beef cattle farmers N = 393).**

| Variables | Coefficients (β) | Std. Error |
|---|---|---|
| Age of a farmers | 0.197* | 0.035 |
| Education level of a farmer | -0.003 | 0.431 |
| Access to market information | -0.432* | 0.915 |
| Distance to market | 0.148* | 0.220 |
| Household size | -0.027 | 0.060 |
| Beef cattle herd size | 2.703* | 0.114 |
| Farming Experience | -0.127* | 0.031 |
| Off-farm income | 0.260* | 0.000 |
| Access to veterinary services | -0.095 | 1.034 |
| Access to credits | -0.048 | 1.254 |
| Grazing land owned | 0.047 | 0.004 |
| Price of beef cattle(market price) | 0.110* | 0.000 |
| Practicing beef cattle fattening | 0.399* | 1.590 |
| Cooperative membership | -0.088* | 0.615 |
| Cows owned | 1.023* | 0.114 |
| Bulls owned | 0.476* | 0.124 |
| Steers-oxen owned | 0.222* | 0.123 |
| Heifers owned | 0.512* | 0.111 |
| Local breed beef cattle | 0.809* | 0.053 |
| Constant | -5.165 | 2.375 |
| R Squared ($R^2$) | 0.792(79.2%) | |
| Adjusted R squared (Adj.$R^2$) | 0.780(78.0%) | |

*Indicate significance level at 5% (P < 0.05).

independent variables account for approximately 79.2% of the change in the total volume of beef cattle sold in the market. The model was estimated using SPSS v.22, and the model was a good fit and significant at $P < 0.05$.

Holding other factors constant, age of the beef cattle farmers had a positive effect on the volume of beef cattle sold in the market and was statistically significant at the 5% significance level. The positive effect indicates that as the farmer's age increases, the volume of beef cattle sold in the market increases. Age of a farmer was closely associated with the decision on what volume of beef cattle to offer for sale. This may also suggest the cumulative effect of cattle over time and acquired experience in beef cattle production and marketing among older farmers. These results correspond with the study by Randela et al. [25], who found a positive association between age and the extent of market participation.

Market information leads to improved beef cattle productivity, hence increased surplus beef cattle for sale in the market [8]. This study, however, revealed that access to some market information negatively influenced the volume of beef cattle sold in the market, and was statistically significant at the 5% significance level. This implies that access to inaccurate market information discouraged beef cattle farmers from offering more beef cattle for sale.

The distance from home to the beef cattle market had a positive impact on the volume of beef cattle sold in the market and was statistically significant at the 5% significance level. According to [26], travelling long distances in search of beef cattle markets consumes time and increases transportation costs, thus increasing marketing transaction costs. Higher transaction costs hinder beef cattle farmers' market participation. In rural areas however, the farther the market location the higher and more profitable the price for beef cattle. Distance to markets motivates farmers to offer more beef cattle for sale. Market infrastructures in the proximity need to be improved to facilitate reliable pricing methods and market information.

As expected, beef cattle herd size had a positive effect on the volume of beef cattle sold in the market and was statistically significant at a 5% significance level. The results indicate that as beef cattle herd size increased, the volume of beef cattle offered for sale also increased. This study is in line with the previous studies by [18, 23] which observe that beef cattle herd size is directly linked to the increases in the beef cattle for sale. This indicates that the larger the beef cattle herd size, the more likely it is to enter the market by selling more beef cattle [18].

Beef cattle farming experience indicated a significant negative effect on the volume of beef cattle sold in the market. This result does not correspond with the studies by [8, 18, 27–29], which found a positive correlation between the increase in farming experience and market participation rate. Farming experience captures the influence of social networks and links accumulated over time to enhance the search for potential customers [23]. Moreover, knowledge and production techniques are acquired through farming experience and are known to be useful in production and marketing activities [29]; however, experienced beef cattle farmers in rural areas are inefficient in producing surplus beef cattle, hence decreasing the volume of beef cattle to be offered for sale.

The off-farm income factor had a positive effect on the volume of beef cattle sold in the market and was statistically significant at a 5% significance level. This implies that keeping other factors constant, beef cattle farmers with a high off-farm income are likely to offer more volume of beef cattle for sale. This study corresponds with the studies by [18, 30], who reported that higher off-farm income motivates beef cattle farmers to participate more in the market by offering more beef cattle for sale, nevertheless, this finding is not in line with the study by [31], who reported a negative correction between off-farm income and the decision to engage in beef cattle marketing.

Price of beef cattle indicated a positive impact on the volume of beef cattle sold in the market and it was statistically significant at a 5% significance level. This indicates that when the

price of beef cattle goes up, beef cattle farmers increase the volume of beef cattle for sale. Higher prices act as an incentive for beef cattle farmers to participate in the market by selling more beef cattle. Market prices for beef cattle are the ultimate motivation for beef cattle sellers [18]. Price control on beef cattle market is recommended.

The coefficient for practicing beef cattle fattening (value addition) had a positive effect on the volume of beef cattle sold in the market and was statistically significant at the 5% significance level. Fattening practice adds value to beef cattle and, in turn, increases productivity and market value, which enhances the availability of surplus beef cattle, thereby increasing the volume of beef cattle sales in the market.

Furthermore, being a member of the cooperative was negatively associated with the volume of beef cattle sold in the market and was statistically significant at the 5% significance level. This indicates that cooperatives lack proper strategies to improve farmer's knowledge on beef cattle production and fail to link farmers to social networks to get accurate market information about the beef cattle market. Policies should be formulated to encourage establishment of informed strategic cooperatives, particularly marketing cooperatives.

Addition of beef cattle flock characteristics on the determinants of the volume of beef cattle sold in the market is distinctive to this study. The coefficients of cows, heifers, bulls, and steers-oxen cattle had a positive effect on the volume of beef cattle sold in the market at 5% significance level. Cows, heifers, and bulls are important in the beef cattle herd size multiplication through reproduction. This multiplication motivates beef cattle farmers to offer more beef cattle for sale to the market. Steers-oxen are non-reproductive beef cattle; hence the availability of these stocks encourages beef cattle farmers to offer more beef cattle for sale. Additionally, these findings also showed that though local breed beef cattle are characterized by low rates of growth and productivity compared to crossbreeds (hybrids), its coefficient positively correlated with the volume of beef cattle offered for sale in the market. This is because local breeds are more resistant to diseases, drought and heat stress, hence, their low mortality rate. Low mortality rates results in increases the surplus beef cattle to be offered for sale to the market.

## 4. Conclusion

Market participation is an important way to ensure better income and food security among beef cattle farmers. It also promotes sustainable beef cattle supply to the market. This study examined factors that influenced market participation among beef cattle farmers. More specifically, the study sought to examine factors that influenced the volume of beef cattle sold in markets. The findings in this study indicated that more farmers participated in the market to secure their financial situations, nevertheless, the volume of sales was low. The findings suggest that higher mean values of variables among market participants positively influenced market participation. Furthermore, results in this study showed that price, farming experience, age of a farmer, off-farm income, beef cattle herd size, cooperative membership, distance to market, cattle fattening, and access to market information were significant variables on the volumes of beef cattle sold in the market. Most importantly, the results showed that increase in beef cattle sales mainly depended on the farmer's cattle herd size. Beef cattle farmers are thus encouraged to keep crossbreeds to enhance productivity. Crossbreed cows can increase herd size, which increases the birth rate and quality of beef cattle. Hybrids have the advantage of heterosis that improves beef cattle productivity. These in turn increase the volume of beef cattle sales in the market.

Access to veterinary services should be improved by reducing the consultation costs. Consultations are necessary for improved beef cattle production. Another significant factor to be emphasised is market price for beef cattle. Prices for beef cattle are the ultimate motivation for

beef cattle sellers. The government should take control on beef cattle price to avoid price fluctuations. Additionally, extension services need to be coordinated to provide training on commercial farming, pricing, and beef cattle marketing.

Policy recommendations in this study focus on agricultural and marketing cooperative societies (AMCOS) in Tanzania, which deal with production, processing, transporting, and marketing of various crops. The AMCOS should expand their services to beef cattle farming to enhance beef cattle productivity and marketing. Strong and effective marketing cooperatives should be established to facilitate access to and utilization of farm credits, and market information. Moreover, marketing cooperatives should aim to provide education and training on new technologies and entrepreneurial skills related to beef cattle marketing and production. These strategies will sensitize beef cattle farmers to treat beef cattle farming as business, hence increase market participation. This will further improve beef cattle farmers' livelihood and encourage sustainable supply of beef cattle to both the domestic and international markets.

## Supporting information

**S1 Questionnaire. Questionnaire in English.**
(PDF)

**S2 Questionnaire. Questionnaire in Swahili.**
(PDF)

**S1 Data.**
(SAV)

**S1 File.**
(ZIP)

## Author Contributions

**Conceptualization:** Zhang Yuejie.

**Data curation:** Cornel Anyisile Kibona.

**Formal analysis:** Cornel Anyisile Kibona.

**Funding acquisition:** Zhang Yuejie.

**Investigation:** Cornel Anyisile Kibona.

**Methodology:** Cornel Anyisile Kibona.

**Project administration:** Zhang Yuejie.

**Resources:** Zhang Yuejie.

**Software:** Zhang Yuejie.

**Supervision:** Zhang Yuejie.

**Validation:** Cornel Anyisile Kibona.

**Visualization:** Zhang Yuejie.

**Writing – original draft:** Cornel Anyisile Kibona.

**Writing – review & editing:** Cornel Anyisile Kibona.

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
