## [Decision Letter · Decision Letter 0]

21 Dec 2020

PONE-D-20-32555

Analysis of the Influencing Factors of Traditional Beef Cattle Farmers’ Market Participation in the Meatu District of Simiyu Region, Tanzania.

PLOS ONE

Dear Dr. Kibona,

Thank you for submitting your manuscript to PLOS ONE. After careful consideration, we feel that it has merit but does not fully meet PLOS ONE’s publication criteria as it currently stands. Therefore, we invite you to submit a revised version of the manuscript that addresses the points raised during the review process.

The manuscript was reviewed by two experts in the field, and they have recommended some modifications be made prior to acceptance.

I therefore invite you to make the modifications and write a response to reviewers prior to re-submission.

I wish you the best of luck with your revisions.

Hope you are keeping safe and well in these difficult times.

We look forward to receiving your revised manuscript.

Kind regards,

Simon Clegg, PhD

Academic Editor

PLOS ONE

Reviewers' comments:

Reviewer's Responses to Questions

**Comments to the Author**

1. Is the manuscript technically sound, and do the data support the conclusions?

Reviewer #1: Yes

Reviewer #2: Partly

2. Has the statistical analysis been performed appropriately and rigorously? 

Reviewer #1: Yes

Reviewer #2: Yes

3. Have the authors made all data underlying the findings in their manuscript fully available?

Reviewer #1: Yes

Reviewer #2: No

4. Is the manuscript presented in an intelligible fashion and written in standard English?

Reviewer #1: No

Reviewer #2: No

5. Review Comments to the Author

Reviewer #1: Title: Analysis of the Influencing Factors of Traditional Beef Cattle Farmers’ Market Participation in the Meatu District of Simiyu Region, Tanzania

The topic is interesting. The author tried to identify the determinants of beef cattle production at smallholder level in Tanzania. Outputs of the study can be used as inputs to improve beef cattle productivity in the country. However, the manuscript requires appropriate revision prior to publication. Please have a look at and try to address the detailed comments given in the attached file, inside the manuscript.

In summary:

The English is poor: long and complex sentences are prevalent, grammar issues…thus requires appropriate revision

The introduction part has several redundant words, phrases and sentences.

Literature review part is a repetition of the introduction section, better to remove.

In the materials and methods section (line 183), the respondents selected purposively and randomly. Why purposively? How many were selected purposively? How was the sample size (393) determined? justify

Results are described repeatedly in different sentences.

Draw your conclusion in relation to your objective and hypothesis.

Reviewer #2: This is an interesting article, but it is quite poorly written. There are some very long sentences, sometimes the same word or two words is in the same sentence up to 5 times. Please reword it to make it easier to read, I have tried to flag them up in the review but there are so many of them. Some of the sentences are very long and need splitting

I would usually recommend rejection based on the standard of writing, however I think that the data is important and beneficial to the field, so I have recommended a major revision with the hope that you can tidy the manuscript up and have it written in a much better way to enable simple reading and extrapolation of the data.

Line 21-25- please have spaces between text and numbers in brackets

Your references are in a strange order. It is often better to start at 1 from the intro and work down. A reference management software will do this for you

Line 41- you have sector in a sentence twice- maybe reword

Line 43- these contributions are untapped the full potential …. – this does not make sense. Please reword it.

Lines 56-62- this is difficult to read. Is there any way to simplify this?

Line 63- but are not limited to (add in word)

Line 62-69- again a long difficult to read sentence- can this be shortened?

Line 76- proven effective may sound better? Also why have they not been effective?

Start line with a capital letter after too

Line 75-80- a very long difficult to read sentence which jumps around a lot, can you please simplify?

Please ensure that there is a space between numbers and units- e.g 1.7 tonnes rather than 1.7tonnes throughout

Line 101-102- repetitive from above

I would combine the literature review with the introduction, and shorten it all, as much of it is repetitive.

A map of the study area may be useful?

Line 183- when you say purposefully, do you mean you chose beef farmers over dairy farmers etc? It is a bit misleading as ideally the sample needs to be random- but can be a random selection of beef farmers

Line 184- was a sample size calculation performed? Or is this all beef farmers in the area?

Line 189- it would be nice to have the questionnaire uploaded too

Line 202- not good to start a sentence with an acronym

Line 243- varies how? By how much?

Line 238- is that age of owner or animal?

Line 244-248- another long sentence, consider shortening or breaking

There are a lot of hypotheses in the methodology section which seems odd

Line 316- there are a lot of places where you have several mentions of ‘beef cattle’ in a sentence which makes it a bit difficult to read

I am also not completely sure that all of the methods data needs to be in the methods section.

Line 352- space between 354 and % brackets

Line 360- was 13.4 times higher, or was 13.4 which is higher than that seen in the other group? Please clarify

Line 362- had more beef cattle farming experience

Line 368-369- not sure that this is needed

Line 370- due to …

Line 375- delete of

Line 375-377- repetitive, please revise

Line 384- doesn’t make sense, please reword

Line 384- change most to mostly

I found table 3 very confusing- is it possible to make it clearer and improve the legend so it is easier to understand?

Line 404-406- reword- 100% means nothing, and can remove a participants

Line 406- females

Line 409-410- two formal educations in close proximity

Line 413- two participants in close proximity

Line 419- space after full stop

Line 423- had limited access to farm credit may sound better?

Line 426-429- better to split this sentence into two for ease of reading

Line 438- lower doesn’t need a capital

Table 4- college ed and the gender sections are almost useless as there are none which fit that group

Line 456- 460- another long sentence, please split

Line 477- distances

Line 476- 481- another long sentence which would benefit from being split

Line 489- studies?

Line 540- out doesn’t need a capital

Line 541- space between 354 (90.1%), and line 542

The conclusion is not as much a conclusion, it is more a summary- please rework

6. PLOS authors have the option to publish the peer review history of their article (what does this mean?). If published, this will include your full peer review and any attached files.

Reviewer #1: No

Reviewer #2: No

---

## [Author Response · Author response to Decision Letter 0]

18 Feb 2021

Response to the Reviewers’ Comments 

We thank the editor and reviewers for taking their time to read and give their comprehensive and constructive comments, which have improved our manuscript. Below; we provide a point by point response to your comments and suggestions and how each one has been addressed in the revision.

 Response to the Editors’ Comments

Comment 1: Please ensure that your manuscript meets PLOS ONE's style requirements, including those for file naming.

Response: We thank the editor for reminding us. We have checked and followed all the requirements to make sure our manuscript meets the PLOS ONE standards.

Comment 2: Please include additional information regarding the survey or questionnaire used in the study and ensure that you have provided sufficient details that others could replicate the analyses. For instance, if you developed a questionnaire as part of this study and it is not under a copyright more restrictive than CC-BY, please include a copy, in both the original language and English, as Supporting Information.

Response: We thank the editor for this comment. We have included a copy of the questionnaire, in both the original language (Swahili) and English, as supporting information.

Response to Reviewer 1 Comments

General Comments 

Comment 1: Title: Analysis of the Influencing Factors of Traditional Beef Cattle Farmers’ Market Participation in the Meatu District of Simiyu Region, Tanzania. The topic is interesting. The author tried to identify the determinants of beef cattle production at smallholder level in Tanzania. Outputs of the study can be used as inputs to improve beef cattle productivity in the country. However, the manuscript requires appropriate revision prior to publication. Please have a look at and try to address the detailed comments given in the attached file, inside the manuscript.

Response: We thank the reviewer for these comments. We appreciate the compliment from the reviewer that the outputs of the study can be used as inputs to improve beef cattle productivity in the country. We agree with the reviewer that our manuscript requires appropriate revision prior to publication. We have corrected our manuscript based on the reviewers’ comments. We hope that the current manuscript will satisfy our reviewer.

Comments in Summary from Reviewer 1

Comment 1: The English is poor: long and complex sentences are prevalent, grammar issues, thus requires appropriate revision.

Response: We thank the reviewer for this comment. We are sorry for disappointing our esteemed reviewer with the poor English, long and complex sentences, and grammar issues in our previous manuscript. We have requested a native English speaking colleague and English experts to read and correct the complexity of sentences and English grammar of our manuscript as suggested by the reviewer. We hope the reviewer will be satisfied with the current English grammar and the structure of the sentences.

Comment 2: The introduction part has several redundant words, phrases and sentences

Response: We appreciate this comment from the reviewer. We agree with the reviewer that the introduction part has several redundant words, phrases, and sentences. We checked and removed all the redundant words, phrases, and sentences based on the reviewer’s comments.

Comment 3: Literature review part is a repetition of the introduction section, better to remove.

Response: We thank the reviewer for this comment. We have checked the literature review part and agree with the reviewer to remove it.

Comment 4: In the materials and methods section (line 183), the respondents selected purposively and randomly. Why purposively? How many were selected purposively? How was the sample size (393) determined? Justify.

Response: Thank you very much for this important comment. We are sorry for the mistakes. The respondents were selected randomly. The study targeted traditional beef cattle farmers (N = 24,139) and applied Slovin’s formula to determine a randomly selected sample size of 393 respondents as;

n= N/(1+Ne^2 )= 24,139/(1+24,139 (〖0.05)〗^2 )=393.48 ≈393 

Whereby N is the targeted population size, n is a sample size, and e is the error tolerance level. The number of respondents selected from each village (stratum) was determined by utilizing the percentage proportion as indicated in the current manuscript.

Comment 5: Results are described repeatedly in different sentences.

Response: We thank the reviewer for this comment. We have checked and corrected the description of the results section to avoid repetition 

Comment 6: Draw your conclusion in relation to your objective and hypothesis.

Response: We thank the reviewer for this comment. We have drawn our conclusion in relation to our objective and hypothesis as suggested by reviewers. More comments are welcome. 

Responses to Specific Comments in the annotated PDF from Reviewer 1

-Title Section

Comment 1: A title is not a sentence, thus full stop at the end is not required

Response: We thank the reviewer for this comment. We are sorry for this mistake. We have removed the full stop at the end of the title. In addition, the title has been modified to read “Factors that Influence Market Participation among Traditional Beef Cattle Farmers in the Meatu District of Simiyu Region, Tanzania.”

-Abstract Section 

Comment 2: Please make a space here and next, after the parenthesis.

Response: We thank the reviewer for this comment. We appreciate the help provided by the reviewer in correcting the mistakes in our manuscript. We have added the space as suggested by the reviewer.

Comment 3: This sentence is incomplete and complex. Make it at least four short sentences as follow. In this study, the average age of the interviewees was 54.52 years with a family size of 13.41. On average, the respondents have about 24.14 years (not 25.04) farming experience. The average off-farm income was found to be 1228 US$. Among the respondents, about 63.8% have access to market information. While the average cattle herd size is 54.46 heads, the average grazing land size is about 39.88 ha per a respondent. 

Response: We appreciate the comment from the reviewer. We are sorry for the incomplete and complex sentence. We have followed the suggestions from the reviewer to rewrite the sentence into four short sentences.

Comment 4: Note: cows, bulls, heifers; these are components of the herd and separate report is not important in the abstract.

Response: We thank the reviewer for this comment. We appreciate the help provided by the reviewer in correcting the abstract section. We have removed the herds' components as a separate report in the abstract as suggested by the reviewer in the enclosed pdf-file.

-Introduction Section

Comment 5: introduction is the first parts of the manuscript that require literature citation. Thus, numbering of the reference should start from 1 here and proceed accordingly.

 Response: We thank the reviewer for this comment. We have followed the suggestions from the reviewer using the enclosed pdf-file to renumber the reference in the introduction part starting from 1 and proceeding accordingly. However some references have been cited more than once, thus the reference numbers may appear several times.

Comment 6: good to say The beef cattle production in Tanzania.... because the word sector is repeated twice in a sentence. 

Response: We appreciate the comment from the reviewer. We have followed the suggestions from the reviewer using the enclosed pdf-file to rewrite the sentence to read as Beef cattle production in Tanzania, because the word sector is repeated twice in a sentence.

Comment 7: These contributions are untapped the full potential" this phrase is not clear or seems incomplete. Please make a meaningful sentence before stating the causes. i,e. due to the limited 

Response: We thank the reviewer for this comment. We are very sorry for the incomplete and unclear sentences. We have corrected the phrase to make a meaningful sentence.

Comment 8: long and complex sentence! make it short and informative! 

Response: We thank the reviewer for this comment. We are very sorry for the long and complex sentence. We have corrected the sentence and make it short and informative based on the reviewer's comment.

Comment 9: About 100 words in one sentence! please rewrite, and make short and clear sentences!

Response We thank the reviewer for this comment. We are sorry for the long and complex sentence. We have rewritten the sentences based on the comment from the reviewer in the enclosed PDF file.

Comment 10: beef cattle and beef... not necessary!. Enough to say ... the increasing demand for meat in both the domestic and international

Response: We thank the reviewer for this comment. We have rephrased the sentence based on the comments from the reviewer.

Comment 11: Provided; the efforts have not provided effective:” this is incomplete! delete it or modify it

Response: We thank the reviewer for this comment. We are sorry for the incomplete sentence and for confusing the reviewer. We have agreed to delete the sentence as suggested by the reviewer to avoid confusion.

Comment 12: Close the sentence here!

Response: We thank the reviewer for this comment. We have corrected the sentence based on the comment from the reviewer in the enclosed PDF file.

Comment 13: Already mentioned around line 48, thus delete it. 

Response: We thank the reviewer for this observation. We checked and deleted the phrase as suggested by the reviewer in the enclosed PDF file.

Comment 14: Better to delete

Response: We thank the reviewer for this comment. We have agreed with the reviewer to delete the phrase to avoid confusion.

Comment 15: Better to delete, already mentioned

Response: We thank the reviewer for the observation. We agree with the reviewer. We have deleted the phrase to avoid repetition. 

Literature Review Section

Comment 16: better to exclude the literature review part! because the introduction part has already several literature sources with the same information mentioned here. 

Response: We thank the reviewer for this comment. We have excluded the literature review part as advised by the reviewer in the enclosed PDF file. 

Materials and Methods Section

Comment 17: why purposely? needs justification. Because, to be representative, respondents need to be selected at a random. 

Response: We thank the reviewer for this comment. We are sorry for the unforeseen mistake. The respondents were randomly selected.

Comment 18: How was the sample size determined? 

Response: We thank the reviewer for this comment. The study targeted traditional beef cattle farmers (N = 24,139) and applied Slovin’s formula to determine a randomly selected sample size of 393 respondents as;

n= N/(1+Ne^2 )= 24,139/(1+24,139 (〖0.05)〗^2 )=393.48 ≈393 

Whereby N is the targeted population size, n is a sample size, and e is the error tolerance level. The number of respondents selected from each village (stratum) was determined by utilizing the percentage proportion as indicated in our current manuscript.

Comment 19: Delete the blank space

Response: We thank the reviewer for this comment. The blank space has been deleted.

Comment 20: The OLS is...

Response: Thank you for the observation. We have corrected the phrase to start with ‘The OLS is’ as suggested by the reviewer in the enclosed PDF file.

Comment 21: Incomplete or rewrite it as follow: These variables are assumed to have influence on the volume of beef cattle sold in the market

Response: We thank the reviewer for this comment. We are sorry for the incomplete sentence and for confusing the reviewer. We have deleted the rephrase to avoid confusion.

Comment 22: close the sentence here, and start new sentence as follow: According to Randela et al. [32], the farmers' age and their decision to increase the volume of beef cattle sales have positive correlation although Kgosikoma and Malope [31] found a negative relationship.

Response: We thank the reviewer for this comment. We have closed the sentence and started a new sentence as suggested by the reviewer in the enclosed PDF file. However, this section has been deleted based on the reviewer’s suggestion.

Results and Discussion Section

Comment 23: Among the 393 sampled beef cattle farmers, about 90.1% (354) had participated....

Response: We thank the reviewer for this comment. We have rephrased the sentence to read, “Among the 393 sampled beef cattle farmers, about 90.1% (354) had participated”, as suggested by the reviewer in the enclosed PDF file.

Comment 24: The farmers are dependent on .

Response: We thank the reviewer for the comment. The highlighted section has been rewritten as suggested by the reviewer.

Comment 25: 13.41, which is higher than the 10.44 obtained for non-market....

Response: We thank the reviewer for the comment. We have rephrased the sentence to read, “The average household size among market participants was 13.41, which is higher than the 10.44 obtained for non-market participants” for clarity as suggested by the reviewer in the enclosed PDF file.

Comment 26: Redundant!

Response: We thank the reviewer for this important observation. We have deleted the phrase accordingly.

Comment 27 Due to the

Response: We thank the reviewer for this comment. We are sorry for the mistake. The missing words have been inserted as suggested by the reviewer.

Comment 28: Space before currency

Response: We thank the reviewer for this comment. The space before currency has been added as suggested by the reviewer in the enclosed PDF file.

Comment 29: delete

Response: We thank the reviewer for the comment. We agree with the reviewer. The highlighted sentence in the enclosed PDF file has been deleted.

Comment 30: Delete "of" and rewrite the sentence as follow: The mean grazing land owned by market participant farmers was 39.88 ha, while the corresponding value for non-market participant was 24.38ha.

Response: We thank the reviewer for this comment. We have deleted the word ‘of’ and rewritten the sentence as suggested by the reviewer in the enclosed PDF file.

Comment 31: abbreviated as ha in the abstract.

Response: We thank the reviewer for the comment. We agree with the reviewer. We have abbreviated the word hectares as ‘ha” as in the abstract to avoid confusion.

Comment 32: mostly.

Response: We thank the reviewer for the comment. We have changed the word ‘most’to mostly as suggested by the reviewer in the enclosed PDF file.

Comment 33: Highlighted word (marketable surplus)

Response: We thank the reviewer for the comment. We have rewritten the word marketable surplus to mean surplus beef cattle.

Comment 34: May be better to say “... the higher average values of these variables.

Response: We thank the reviewer for the comment. We have agreed, the sentence has been rephrased as suggested by the reviewer in the enclosed PDF file.

Comment 35: This is confusing! you have not reported crossbreeds and exotic breeds, thus the cattle you listed are locals. Why you mentioned the phrase local breeds owned? again the average should be the same with beef cattle herd size. Or make it clear. Local breed animal (cattle) is more informative than "local breeds owned". It looks like that you are counting the number of breeds that are local.

Response 36: We thank the reviewer for these comments. We are sorry for the mistakes and confusion to the reviewer. To avoid confusion, we have deleted the phrases local breeds and crossbreeds owned. This indicates that all the beef cattle mentioned here are local. 

Comment 37: delete 

Response: We thank the reviewer for this comment. The highlighted word participants have been deleted throughout the manuscript as suggested by the reviewer.

Comment 38: Rewrite it

Response: We thank the reviewer for this comment. The highlighted section has been rewritten as suggested by the reviewer in the enclosed PDF file.

Comment 39: highlighted text

Response: We thank the reviewer for the comment. The highlighted words have been rewritten as suggested by the reviewer to read “the remaining 28.8% and 20.5% of market and non-market participants, respectively, had no formal education” to make it clear.

Comment 40: Rewrite

Response: We thank the reviewer for the comment. The highlighted section has been rewritten as suggested by the reviewer.

Comment 41: Had poor access to farm credits.

Response: We thank the reviewer. We agree with the reviewer. We have rephrased the sentence to read “had poor access to farm credits” as suggested by the reviewer.

Comment 42: Close the sentence here.

Response: We thank the reviewer for the comment. We have closed the sentence as suggested by the reviewer in the enclosed PDF file.

Comment 43: Among non-market participants

Response: We thank the reviewer for this comment. We have rephrased the sentence as suggested by the reviewer. 

Comment 44: lower

Response: We thank the reviewer for this comment. We are sorry for the typing error. We have corrected the capital letter L to read as small letter in the word “lower”.

Comment 45: The dote (.)

Response: We thank the reviewer for this close observation. We are sorry for the typing error. We have inserted the full stop and deleted the coma.

-Conclusion Section

Comment 46: Your conclusion should adhere to the key findings of the study i.e. the hypothesis or objectives. It should not be another results and discussion

Response: We thank the reviewer for the important comment. We have drawn our conclusion in relation to our key findings, objective, and hypothesis as suggested by reviewers. More comments are welcome.

Comment 47: Already abbreviated 

Response: We thank the reviewer for this comment. We agree with reviewer. We have deleted the phrase since it is already abbreviated.

Comment 48: Not part of this study.

Response: We thank the reviewer for this comment. We checked and deleted the highlighted phrase since it is not directly the part of this study.

Response to Reviewer 2

General Comments

Comment 1: This is an interesting article, but it is quite poorly written. There are some very long sentences, sometimes the same word or two words is in the same sentence up to 5 times. Please reword it to make it easier to read, I have tried to flag them up in the review but there are so many of them. Some of the sentences are very long and need splitting.

Response: We thank the reviewer for seeing the useful information contained in our manuscript and consider it as an interesting article despite its weakness. We are sorry for disappointing our esteemed reviewer for the poor English, long and complex sentences, and grammar issues in the previous version of our manuscript. We have improved our manuscript based on the reviewers’ comments. We hope the reviewer will find it more useful. 

Comment 2: I would usually recommend rejection based on the standard of writing; however I think that the data is important and beneficial to the field, so I have recommended a major revision with the hope that you can tidy the manuscript up and have it written in a much better way to enable simple reading and extrapolation of the data.

Response: We thank the reviewer for this comment. We are sorry for the shortcomings in our previous manuscript. We appreciate the compliment from the reviewer that the data is important and beneficial to the field. We diligently and carefully corrected our manuscript in a much better way to enable simple reading and extrapolation of the data based on the valuable comments given by the reviewer. We kindly request you to reconsider it for publication. 

Specific Comments from Reviewer 2

Abstract Section

Comment 1: Line 21-25- please have spaces between text and numbers in brackets

Response: We thank the reviewer for this comment. The spaces between text and numbers in blackest have been added. However the section has been rephrased based on the reviewer comment.

Introduction Section

Comment 2: Your references are in a strange order. It is often better to start at 1 from the intro and work down. Reference management software will do this for you.

Response: We thank the reviewer for this important comment. We have followed the suggestions from the reviewer to renumber the reference in the introduction part starting from 1 and proceeding accordingly. However some references have been cited more than once, thus the reference numbers may repeat.

Comment 3: Line 41- you have sector in a sentence twice- maybe reword. 

Response: We thank the reviewer for yet another important reminder. We have corrected the whole manuscript to ensure that the word sector or other words do not appear twice in a sentence. The word sector in a given phrase has been changed to production to read, “Beef cattle production in Tanzania is one of the major agricultural production sectors….

Comment 4: Line 43- these contributions are untapped the full potential …. – this does not make sense. Please reword it

Response: We thank the reviewer for this comment. We are very sorry for the incomplete and unclear sentences. We have rephrased the sentence; these contributions are untapped the full potential to read, “Despite its potential for economic development, the traditional beef cattle sector has thinly been developed, partly due to the limited commercialization (market participation).

Comment 5: Lines 56-62- this is difficult to read. Is there any way to simplify this?

Response: We thank the reviewer for this comment. We have simplified and shortened the phrase to make it clear. We hope it is clear now.

Comment 6: Line 63- but are not limited to (add in word)

Response: We thank the reviewer for this comment. We are sorry for confusing the reviewer. We have checked and deleted this phrase to avoid confusion.

Comment 7: Line 62-69- again a long difficult to read sentence- can this be shortened?

Response: We thank the reviewer for this comment. We are sorry for a long difficult to read sentence. We have rephrased, and make short based on the comment from the reviewer.

Comment 8: Line 76- proven effective may sound better? Also why have they not been effective? 

Response: We thank the reviewer for this comment. We have agreed to rephrase the sentence following the suggestion given by the reviewer. However this phrase has been deleted as suggested by the reviewer in the enclosed PDF file to avoid confusion.

Comment 9: Start line with a capital letter after too.

Response: We thank the reviewer for this comment. The phrase has been modified and each line has been started by capital letter.

Comment 10: Line 75-80- a very long difficult to read sentence which jumps around a lot, can you please simplify?

Response: We thank the reviewer for this comment. We are sorry for a very long difficult to read sentence which jumps around a lot. We have rephrased, and make short and clear sentence based on the comment from the reviewer

Comment 11: Please ensure that there is a space between numbers and units- e.g 1.7 tonnes rather than 1.7tonnes throughout. 

Response: We thank the reviewer for this comment. The space between numbers and units has been added throughout the current manuscript.

Comment 12: Line 101-102- repetitive from above

Response: We thank the reviewer for this observation. We checked and deleted the phrase since it is repetitive as commented by the reviewer.

Literature Review Section

Comment 13: I would combine the literature review with the introduction, and shorten it all, as much of it is repetitive

Response: We thank the reviewer for this important comment. We have agreed to combine the literature review with introduction and shortened it; however a large part of the literature review has been deleted to avoid repetition as suggested by the reviewer.

Materials and Methods Section

Comment 14: A map of the study area may be useful?

Response: We thank the reviewer for this important suggestion. We acknowledge the suggestion by the reviewer wishing inclusion of a physical map in our article. However, given the complexities in obtaining available technically and locally constructed maps of specific areas in the country, the map could not be included. We hope the omission of the map does not in any manner devalue the quality of geographical descriptions of the study location.

Comment 15: Line 183- when you say purposefully, do you mean you chose beef farmers over dairy farmers etc? It is a bit misleading as ideally the sample needs to be random- but can be a random selection of beef farmers

Response: We thank the reviewer for this comment. We are sorry for the unforeseen mistake and misleading the reviewer. The respondents (beef cattle farmers) were randomly selected 

Comment 16: Was a sample size calculation performed? Or is this all beef farmers in the area?

Response: We thank the reviewer for this comment. A sample size calculation was performed. The study targeted traditional beef cattle farmers (N = 24,139) and applied Slovin’s formula to determine a randomly selected sample size of 393 respondents as.

n= N/(1+Ne^2 )= 24,139/(1+24,139 (〖0.05)〗^2 )=393.48 ≈393 

Whereby N is the targeted population size, n is a sample size, and e is the error tolerance level. The number of respondents selected from each village (stratum) was determined by utilizing the percentage proportion as indicated in the current new manuscript.

Comment 17: Line 189- it would be nice to have the questionnaire uploaded too

Response: We thank the reviewer for this comment. A questionnaire has been uploaded as supplementary information.

Comment 18: Line 202- not good to start a sentence with an acronym

Response: We thank the reviewer for this comment. We have corrected the phrase to start with ‘The OLS is’ as suggested by the reviewer.

Comment 19: Line 243- varies how? By how much?

Response: We thank the reviewer for this comment. We are sorry for the incomplete sentence and for confusing the reviewer. We have deleted the phrase to avoid confusion.

Comment 20: Line 238- is that age of owner or animal?

Response: We thank the reviewer for this comment. We have corrected in our new manuscript to indicate the age as the age of a beef cattle farmer.

Comment 21: Line 244-248- another long sentence, consider shortening or breaking

Response: We thank the reviewer for this comment. We are sorry for the long sentence. We have shortened the sentence to make it clear and easy to read. However, this section has been deleted, as suggested by the reviewer.

Comment 22: There are a lot of hypotheses in the methodology section which seems odd

Response: We thank the reviewer for this comment. We are sorry for introducing many hypotheses in the methodology section, which seems odd; however, we aimed to justify the validity of the variables selected for this study. We have deleted this section to avoid too many hypotheses.

Comment 23: Line 316- there are a lot of places where you have several mentions of ‘beef cattle’ in a sentence which makes it a bit difficult to read.

Response: We thank the reviewer for this comment. This section has been deleted to avoid confusion.

Comment 24: I am also not completely sure that all of the methods data needs to be in the methods section.

Response: We thank the reviewer for this comment. Some of the methods data (line 244 - 320) in the methods section has been deleted.

Results and Discussion Section

Comment 25: Line 352- space between 354 and % brackets.

Response: We thank the reviewer for this comment. The space between 354 and % brackets has been added as suggested by the reviewer.

Comment 26: Line 360- was 13.4 times higher, or was 13.4 which is higher than that seen in the other group? Please clarify.

Response: We thank the reviewer for this comment. We have corrected the phrase to read “the average household size among market participants was 13.41, which is higher than the 10.44 obtained for non-market participants”.

Comment 27: Line 362- had more beef cattle farming experience

Response: We thank the reviewer for this comment. We have rewritten the phrase to read “market participants had 25.04 years of farming experience, which is higher than the 18.74 years obtained for non-market participants”.

Comment 28: Line 368-369- not sure that this is needed

Response: We thank the reviewer for this comment. We have checked and agreed with the reviewer. The highlighted phrase has been deleted. 

Comment 29: Line 370- due to …

Response: We thank the reviewer for this comment. We are sorry for the mistake. We have added the missing words, as suggested by the reviewer.

Comment 30: Line 375- delete of

Response: We thank the reviewer for this comment. The word “of” has been deleted as suggested by the reviewer.

Comment 31: Line 375-377- repetitive, please revise.

Response: We thank the reviewer for this comment. We have revised the sentence as follows: The mean grazing land owned by market participant farmers was 39.88 ha, while the corresponding value for the non-market participant was 24.38ha.

Comment 32: Line 384- doesn’t make sense, please reword.

Response: We thank the reviewer for this comment. We have rewritten the phrase to read “on average, steers owned by beef cattle farmers were found to be high among non-market participants, thus affected the market participation”.

Comment 33: Line 384- change most to mostly

Response: We thank the reviewer for this comment. We have changed the word “most” to mostly.

Comment 34: I found table 3 very confusing- is it possible to make it clearer and improve the legend so it is easier to understand?

Response: We thank the reviewer for this comment. We are sorry for the mistakes and the confusion to the reviewer. We have improved the legend by deleting the phrases local breeds and crossbreeds owned. It means that all the beef cattle mentioned here are local.

Comment 35: Line 404-406- reword- 100% means nothing, and can remove a participants

Response: We thank the reviewer for this comment. The highlighted word participant has been deleted throughout the manuscript as suggested by the reviewer. In addition, the phrase “100% of males dominate the beef cattle farming” has been rewritten to read a traditional beef cattle farming was dominated only by men.

Comment 36: Line 406- females.

Response: We thank the reviewer for the comment. The word female has been changed to females as suggested by the reviewer.

Comment 37: Line 409-410- two formal educations in close proximity

Response: We thank the reviewer for the comment. The phrases have been rewritten to read “the remaining 28.8% and 20.5% of the market and non-market participants, respectively, had no formal education”.

Comment 38: Line 413- two participants in close proximity.

Response: We thank the reviewer for this comment. We have rephrased the sentence to read, “63.8% and 38.5% of the market and non-market participants, respectively, had access to market information”. We have deleted the word participants to avoid repetition.

Comment 39: Line 419- space after full stop.

Response: We thank the reviewer for this comment. We have added the space after full stop.

Comment 40: Line 423- had limited access to farm credit may sound better?

Response: We thank the reviewer for this comment. We have rephrased the sentence to read “had poor access to farm credits”.

Comment 41: Line 426-429- better to split this sentence into two for ease of reading

Response: We thank the reviewer for this comment. We have agreed with the reviewer to split the sentence into two for ease of reading as indicated in our new manuscript with track changes.

Comment 42: Line 438- lower doesn’t need a capital.

Response: We thank the reviewer for this comment. We are sorry for the typing error. We have corrected the capital letter L to read as small letter in the word “lower”.

Comment 43: Table 4- college ed and the gender sections are almost useless as there are none which fit that group.

Response: We thank the reviewer for this comment. We have deleted the college ed and gender sections, particularly females section as suggested by the reviewer. More suggestions are welcome.

Comment 44: Line 456- 460- another long sentence, please split.

Response: We thank the reviewer for this comment. We are sorry for the long sentence. We have decided to remove the section to avoid repetition.

Comment 45: Distances.

Response: We thank the reviewer for this comment. We are sorry for the typing error. We have corrected the word “distance” to distances. 

Comment 46: Line 476- 481- another long sentence which would benefit from being split.

Response 47: We thank the reviewer for this comment. We are sorry for the long sentence. We agreed with the reviewer to split the sentence into three short sentences for easy reading. 

Comment 48: Line 489- studies?

Response: We thank the reviewer for the comment. The word study has been changed to studies as suggested by the reviewer.

Comment 49: Line 540- out doesn’t need a capital.

Response: We thank the reviewer for this comment. We are sorry for the typing error. We have removed a capital letter from the word “out” as suggested by the reviewer.

Comment 50: Line 541- space between 354 (90.1%), and line 542.

Response: We thank the reviewer for this comment. This section has been rephrased to improve the conclusion section as suggested by the reviewers.

Conclusion Section

Comment 51: The conclusion is not as much a conclusion, it is more a summary- please rework

Response: We thank the reviewer for the important comment. We have drawn our conclusion in relation to our key findings, objective, and hypothesis. We hope this will be clear to our reviewer.

---

## [Decision Letter · Decision Letter 1]

2 Mar 2021

Factors that Influence Market Participation among Traditional Beef Cattle Farmers in the Meatu District of Simiyu Region, Tanzania

PONE-D-20-32555R1

Dear Dr. Kibona,

We’re pleased to inform you that your manuscript has been judged scientifically suitable for publication and will be formally accepted for publication once it meets all outstanding technical requirements.

Kind regards,

Simon Clegg, PhD

Academic Editor

PLOS ONE

Additional Editor Comments:

Many thanks for resubmitting your manuscript to PLOS One

As you have addressed all the comments and the manuscript reads well, I have recommended it for publication

You should hear from the Editorial Office shortly.

It was a pleasure working with you and I wish you the best of luck for your future research

Hope you are keeping safe and well in these difficult times

Thanks

Simon

Reviewers' comments:

Reviewer's Responses to Questions

**Comments to the Author**

1. If the authors have adequately addressed your comments raised in a previous round of review and you feel that this manuscript is now acceptable for publication, you may indicate that here to bypass the “Comments to the Author” section, enter your conflict of interest statement in the “Confidential to Editor” section, and submit your "Accept" recommendation.

Reviewer #1: All comments have been addressed

2. Is the manuscript technically sound, and do the data support the conclusions?

Reviewer #1: Yes

3. Has the statistical analysis been performed appropriately and rigorously? 

Reviewer #1: Yes

4. Have the authors made all data underlying the findings in their manuscript fully available?

Reviewer #1: (No Response)

5. Is the manuscript presented in an intelligible fashion and written in standard English?

Reviewer #1: Yes

6. Review Comments to the Author

Reviewer #1: The comments are addressed adequately. The present version of the manuscript is greatly improved. I recommend the manuscript for publication at PLOS ONE.

7. PLOS authors have the option to publish the peer review history of their article (what does this mean?). If published, this will include your full peer review and any attached files.

Reviewer #1: No

---

## [Editor Report · Acceptance letter]

25 Mar 2021

PONE-D-20-32555R1 

Factors that Influence Market Participation among Traditional Beef Cattle Farmers in the Meatu District of Simiyu Region, Tanzania 

Dear Dr. Yuejie:

I'm pleased to inform you that your manuscript has been deemed suitable for publication in PLOS ONE. Congratulations! Your manuscript is now with our production department. 

Kind regards, 

on behalf of

Dr. Simon Clegg 

Academic Editor

PLOS ONE